# Altered light induced EGR1 expression in the SCN of PACAP deficient mice

Casper Schwartz Riedel[ID], Birgitte Georg, Jan Fahrenkrug, Jens Hannibal[ID]*

Department of Clinical Biochemistry, Faculty of Health Sciences, Bispebjerg Hospital, University of Copenhagen, Copenhagen NV, Denmark

* j.hannibal@dadlnet.dk

**Data Availability Statement:** All relevant data are within the paper and its Supporting Information files.

**Funding:** The study was supported by the Danish Biotechnology Center for Cellular Communication.

## Abstract

The brain's biological clock is located in the suprachiasmatic nucleus (SCN) of the hypothalamus and generates circadian rhythms in physiology and behavior. The circadian clock needs daily adjustment by light to stay synchronized (entrained) with the astronomical 24 h light/dark cycle. Light entrainment occurs via melanopsin expressing retinal ganglion cells (mRGCs) and two neurotransmitters of the retinohypothalamic tract (RHT), PACAP and glutamate, which transmit light information to the SCN neurons. In SCN neurons, light signaling involves the immediate-early genes *Fos*, *Egr1* and the clock genes *Per1* and *Per2*. In this study, we used PACAP deficient mice to evaluate PACAP's role in light induced gene expression of EGR1 in SCN neurons during early (ZT17) and late (ZT23) subjective night at high (300 lux) and low (10 lux) white light exposure. We found significantly lower levels of both EGR1 mRNA and protein in the SCN in PACAP deficient mice compared to wild type mice at early subjective night (ZT17) exposed to low but not high light intensity. No difference was found between the two genotypes at late night (ZT23) at neither light intensities. In conclusion, light mediated EGR1 induction in SCN neurons at early night at low light intensities is dependent of PACAP signaling. A role of PACAP in shaping synaptic plasticity during light stimulation at night is discussed.

## Introduction

Circadian rhythms in physiology and behaviour are in mammals generated from the suprachiasmatic nucleus (SCN), located in the ventral hypothalamus. The SCN consists of approximately 20,000 neurons and most of these neurons express a circadian clock, which is synchronized by internal neuronal signals to a coherent rhythm close to 24 h cycles [1, 2]. The molecular machinery of the biological clock consists of a group of clock genes, which by complex feedback interactions between the clock proteins drive circadian rhythmicity within neurons of the SCN (Mohawk and Takahashi, 2011; Mohawk et al., 2012). The endogenous period length of the circadian clock deviates slightly from 24 h and neurons of the SCN are therefore daily synchronized (entrained) to the astronomical day by light, which is the most important "zeitgeber" for entrainment (Golombek and Rosenstein, 2010). In mammals, light information for entrainment is transmitted from intrinsically photosensitive retinal ganglion cells

**Competing interests:** The authors have declared that no competing interests exist.

(ipRGCs) containing the photoreceptor melanopsin, and by signals from the classical photoreceptors, the rods and cones [3, 4]. The ipRGCs project as the retinohypothalamic tract (RHT) to the brain, including the SCN, and provide light information for the entrainment process, negative masking and the pupillary light reflex [4, 5]. Two neurotransmitters, glutamate and PACAP, several subtypes of glutamate receptors and the PACAP type 1 receptor (PAC1), convey the light signals to the SCN neurons [6–8]. We have previously provided evidence that PACAP mediates synchronization of the SCN neurons via the PAC1 receptor targeting signaling pathways involving the immediate early gene (IEG) *Fos*, and the clock genes *Per1* and *Per2* [7, 9, 10]. Another IEG strongly induced in the SCN by light is NGFI-A, also known as *Egr1* (Early growth response protein 1), *zif*268 and *Krox*-24 [11–14]. Interestingly, PACAP has previously been shown to stimulate EGR1 expression in human neuroblastoma NB1-cells though PAC1 receptor/PKC/MEK1/2 signaling [15], and PACAP induced neuronal differentiation of PC12 cells requires EGR1 signaling pathways [16]. In the present study, we therefore investigated the role of PACAP in light induced expression of EGR1 in the mouse SCN by using quantitative in situ hybridization and immunohistochemistry on wild type and PACAP deficient mice light stimulated early or late subjective night at high (300 lux) or low (10 lux) light intensity.

## Material and methods

### Animals

64 PACAP wild type (+/+) and 64 PACAP deficient (-/-) mice (32 males and 32 females of each genotype, age 8–12 weeks when included in the study) were bred from heterozygote animals in a 129/Sv background [17]. Our strain of PACAP deficient mice was originally provided from Jim Wachek and Chris Colwell [18]. All animals included in the study were maintained in a 12:12 h light/dark (LD) cycle (light on at 6 a.m. = Zeitgeber (ZT) 0, lights off at 6 p.m. = ZT12) and housed in individual cages with food (Altromin 1324; Altromin Spezialfutter, Germany) and water ad libitum unless otherwise stated. Animals were treated according to the principles of Laboratory Animal Care (Law on Animal Experiments in Denmark, publication 382, June 10, 1987) and under Danish Veterinary Authorities (Dyreforsoegstilsynet) license no. 2008/561-1445. The animal research ethics committee (Dyreforsoegstilsynet) granted a formal waiver of ethics approval license no: 2017-15-0201-01364 to Jens Hannibal and thereby approved the study.

### Light experiments

For light-stimulation experiments, animals received a 30 min pulse of white light (>300 lux or 10 lux) at ZT16 or ZT22. In experiments performed at ZT16 using 300 lux, animals were killed after 30, 60 and 120 min after initiation of the light pulse. Here we found equal *Egr1* induction in both groups of animals at all timepoints suggesting similar dynamics in PACAP knockout and wild type mice, and 60 min was hereafter chosen for the remaining experiments (see Fig 1). For in situ hybridization, animals (n = 12–16 in each group, equal number of sexes) receiving either a light pulse or kept as time matched darkness controls, were decapitated in dim red light (<3 lux). For immunohistochemistry, light stimulated animals (n = 4 in each group) were killed and perfusion fixed in dim red light 60 min after the initiation of light exposure in Stefanini's fixative (2% PFA, 15% picric acid in 0.1 M PBS, pH 7.2) followed by immersion fixation in the same fixative overnight. Since we did not find any EGR1 immunoreactivity in the SCN of mice kept in darkness at the two time points (ZT17 and ZT23), comparison of EGR1 immunoreactivity in the SCN was done between light stimulated genotypes.

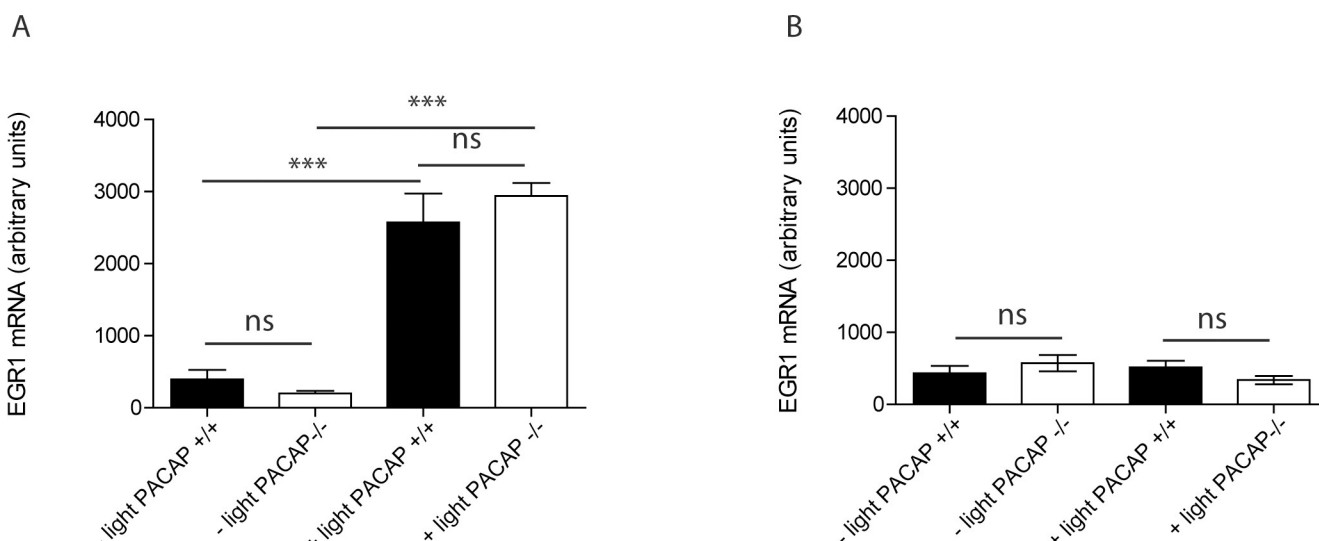

**Fig 1. Egr1 mRNA in SCN of wild type and PACAP deficient mice stimulated with 300 lux at ZT16.** (A) Egr1 mRNA at ZT16:30 (30 min after initiation of the light pulse) and (B) at ZT18 (120 min after initiation of the light pulse). The quantities of Egr1 mRNA (digoxigenin labeled) are presented as group means (± SEM, n = 6–8 animals), and black bars represent wild type mice (controls) and white bars PACAP deficient mice. *** p<0.001.

## Light source and light intensity measurements

White light was delivered by fluorescent tubes placed on top of the cages. The light intensity could be adjusted from 10–900 lux (measured at the top of the cages) via a resistence. The light intensity was set to either 300 (also used during ordinary housing) or 10 lux measured using an Advantest Optical Power meter TQ8210 (MetricTest, Hayward, CA), with measurements determined at setting of 514 nm; 300 lux (115.0 $\mu$W/cm$^2$) and 10 lux (4.3 $\mu$W/cm$^2$), respectively.

## In situ hybridization histochemistry

For detection of *Egr1* mRNA antisense, RNA probes were used. As template nucleotide 1–1978 (BC138615) excised as an EcoRI-fragment from IRCKp5014F0910Q (Source Bioscience, Nottingham, UK) and inserted in the SmaI site of pBluescriptKS+ was used. The resulting plasmid was linearized with HindIII for antisense and with BamHI for the sense probes, and transcription was done using T7 and T3-polymerase, respectively. In situ hybridization was performed using $^{33}$P-labeled probes (ZT17 and ZT23, both light intensities, and a 24 h LD serie, see S1 Material and S1A Fig) as previously described [19] or digoxigenin labeled probes [20] (see S1 Material). Briefly, the brains were cut on a cryostat in 12-$\mu$m-thick coronal sections through the SCN in three series of five slides with 3–4 sections on each slide. From each animal, one gelatin-coated slide from each series representing the rostral, mid, and caudal part of the SCN, respectively, was hybridized with the *Egr1* antisense probe. After hybridization and washing, slides hybridized with $^{32}$P-labeled probes were exposed to Amersham Hyperfilm (Amersham, DK) for 4–7 days. Autoradiograms were photographed by a DC200 camera and Q500MC Image Analysis System version 2.02A; Leica Cambridge, UK). The levels of *Egr1* mRNA in each animal at the rostral, mid, and caudal level of the SCN were quantified (qISH) with Fiji software as described previously [21] by measuring O.D. of the hybridization signals in the bilateral SCN. The measurements were corrected for nonspecific background by subtracting the grayscale values from a neighboring area (the optic chiasma) considered free of positive hybridization. The calculated mean of these measurements from each of the animals

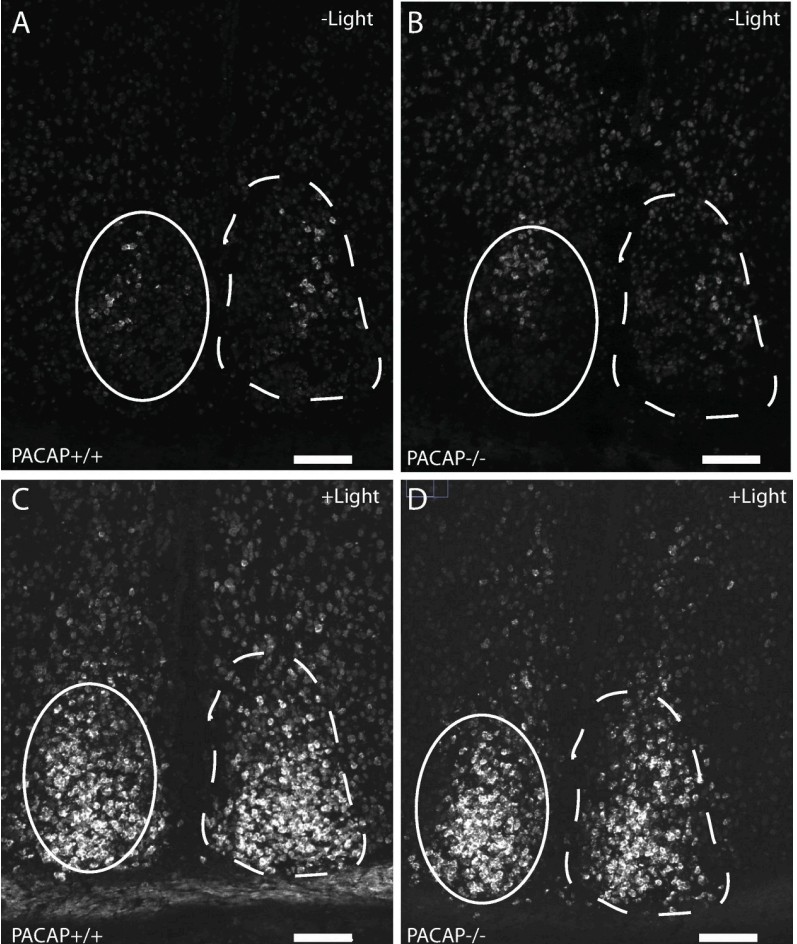

**Fig 2.** *Egr1* mRNA (digoxigenin labeled probe) in the SCN in wild type (A, C) and PACAP deficient mice (B, D) at ZT16:30 (30 min after initiation of a 300 lux light pulse)(C, D) and control animals killed in darkness at ZT16:30 (A,B). Note the small group of cells weakly expressing *Egr1* mRNA in the central part of the central SCN in the dark controls. Throughout the study, *Egr1* mRNA was quantified in the central retinorecipient SCN [5] indicated by solid line. Dashed lines outline the entire mid SCN. Scale bars = 50 μm.

was used to calculate the group mean and SEM. Sections hybridized with digoxigenin labeled probes were analysed and the level of *Egr1* mRNA expression determined by Fiji/ImageJ and described previously [21] (see Fig 2 and S1 Material). Hybridization was routinely performed in parallel using antisense and sense probes on sections from the same animal; no signal was obtained using the sense probe.

## Immunohistochemistry (IHC)

**Tissue preparation.** Before fixation, the animals were anesthetized using a subcutaneous injection of hypnorm and midazolam, 0.1 ml per 10 g of body weight. Hereafter, the animals were transcardially perfused with heparin (15,000 IE/L PBS, pH 7.2) for 3 minutes followed by Stefanini fixative for 15 minutes. All brains were post-fixed overnight in Stefanini fixative and cryoprotected in 30% sucrose prior to freezing (at -20˚C).

Fixed brains were cut in a Leica CM3050S cryostat in 40-μm-thick free-flowing sections, as described previously [22] and stored at −21˚C in cryoprotectant until processed. Brain sections were treated by antigen retrieval solution at 80˚C for 1.5 h (DAKO ChemMate, Glostrup,

Denmark, code No. S 203120 in distilled water, pH 6) and post-fixed in 4% PFA before pro-
cessing for IHC. IHC was performed as previously described [23], using a rabbit anti-Egr-1
antibody (code no: SC-189, diluted 1:10.000, Santa Cruz Biotechnology, Inc.) and ENVI-
SION® secondary antibody complex (Dako (K4002)). Finally, EGR1 was visualized by Alexa
Fluor 488-conjugated tyramide (Molecular Probes, diluted 1:500). Control of EGR1 antibody
specificity was confirmed by staining brain sections of light stimulated EGR1 knockout ani-
mals [24], which showed no staining.

## Photomicrographs

Fluorescent images were obtained using an iMIC confocal microscope (Till Photonics, FEI,
Germany) equipped with appropriate filter settings for detecting DAPI and Cy2/Alexa Fluor
488. The SCN was photographed with the iMIC confocal microscope using filter settings for
Alexa488. Images from 40-micron thick brain sections were obtained as Z-stacks at X20 mag-
nification and consist of 50 digital sections separated in the Z-level by 0.5 μm. Figures were
shown as maximal projections in Fiji software (version 1.47q, NIH, USA) and mounted into
plates using Adobe Illustrator CS5.

## Statistics

Statistics were performed using GraphPad Prism version 5.0. For comparison one-way Anova
followed by Bonferroni's Multiple Comparison Test was used. P < 0.05 was considered statisti-
cally significant.

# Results

## EGR1 mRNA dynamic after light stimulation at night

Light induced EGR1 mRNA expression was significantly increased after a 30 min light pulse
(300 lux) in both genotypes (Fig 1A). No difference was found between the two genotypes. 180
min after the light pulse at ZT16 EGR1 mRNA was almost at the level of the control animals
not receiving the light pulse (Fig 1B) in the two genotypes. At ZT16:30 light induced EGR1
mRNA was located in the retinorecipient areas of the mouse SCN (Fig 2). A smaller number of
neurons in the mid SCN demonstrated a low but distinct expression of EGR1 mRNA in the
control animals (darkness) around ZT16-17 (Fig 2A and 2B).

## Blunted light induced Egr1 mRNA in the SCN of PACAP deficient mice during low but not high light intensity at early night

We next examined the induction of EGR1 mRNA in wild type and PACAP deficient mice
after a 30 min light pulse at early (ZT17) or late (ZT23) night at high (300 lux) and low (10 lux)
light intensities. Theise timepoints were selected based on the observation that Egr1 mRNA
expression was highest 60 min after light stimulation (Fig 1 and Fig 2 and S1 Material). In both
genotypes Egr1 mRNA was significantly induced in the SCN after light stimulation at early
night with 300 lux compared to the controls (Fig 3A). No differences were found between the
wild type and PACAP deficient mice (p = ns) (Fig 3A). 10 lux (low) light intensity, induced
Egr1 mRNA two-fold in the SCN compared to the controls (Fig 3B). PACAP deficient mice
demonstrated, however, a severely blunted response to light stimulation compared to the wild
type animals (p<0.001) (Fig 3B). Immunohistochemical staining of EGR1 protein supported
the Egr1 mRNA findings (Fig 4). At the timepoint examined, EGR1 immunoreactivity was
found widespread in neurons of the core and shell of the SCN (Fig 4). At light intensity of 300
lux no difference was found in EGR1 expression in the SCN between the genotypes (Fig 4A

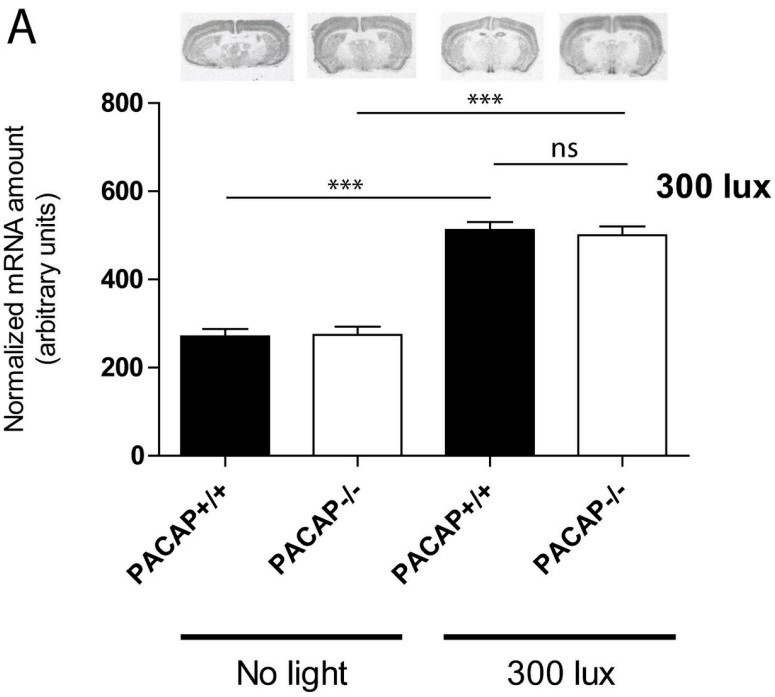

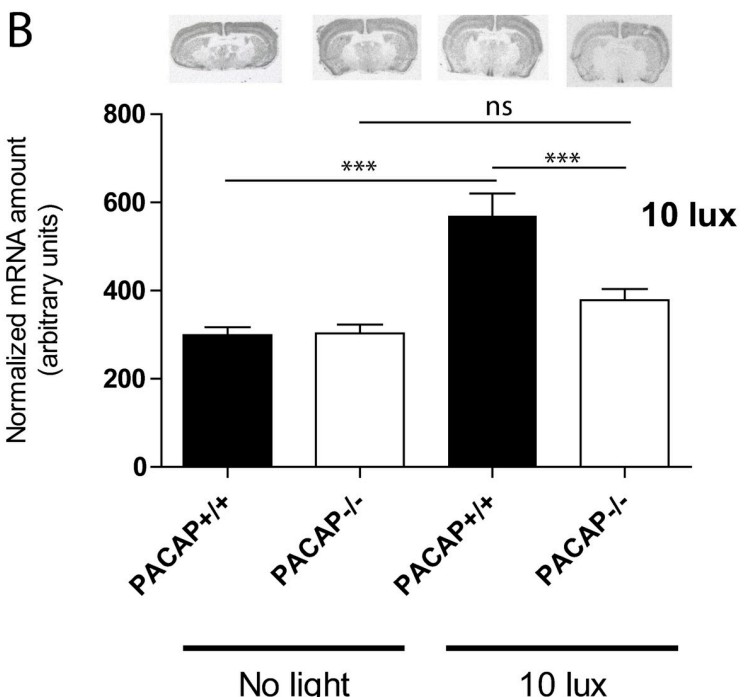

**Fig 3.** EGR1 mRNA (isotop labelled probe) in the SCN in wild type and PACAP deficient mice at early (ZT17) night light stimulated with 300 lux (A) and 10 lux (B) light intensity. EGR1 mRNA amounts are presented as group mean (± SEM, n = 8 animals) and wild type mice (controls) are shown in the black bars and PACAP deficient mice in the white bars. *** $p < 0.001$, ** $p < 0.005$, * $p < 0.05$.

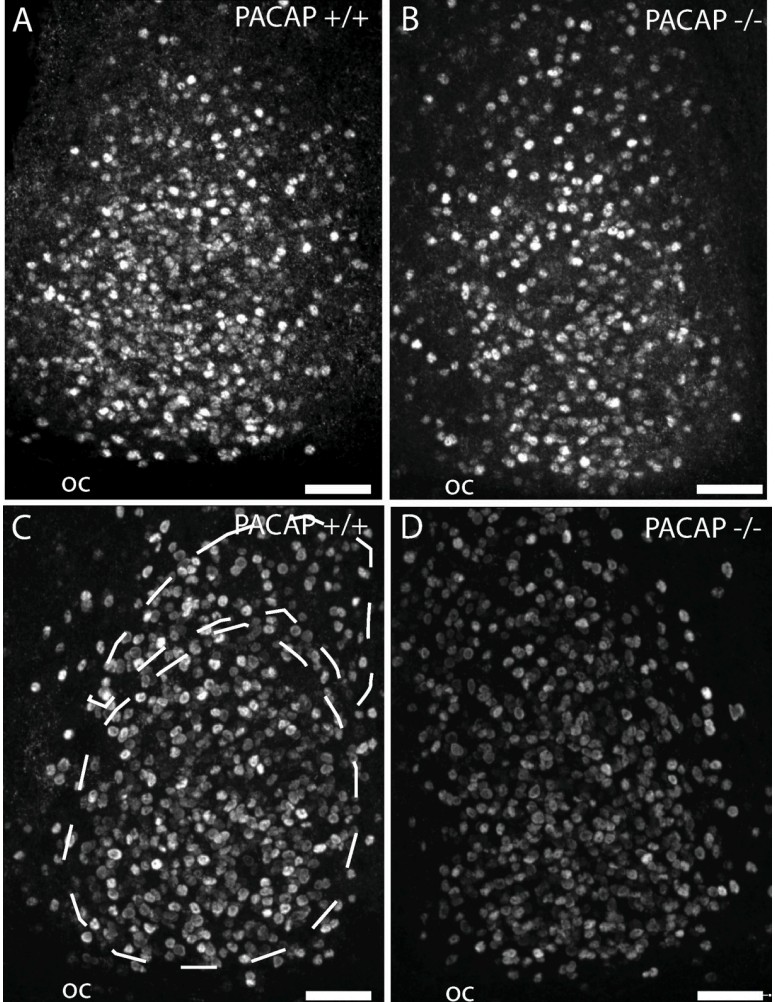

**Fig 4. Immunohistochemistry of coronal sections of the mid SCN stained for EGR1 protein at early night (ZT 17).**
The core and ventral shell (retinorecipient) and shell portion of the mid SCN [41] is indicated in panel C. Animals were fixed 60 min after a 30 min light stimulation at 300 lux (A and B) or 10 lux (C and D). OC, optic chiasm, Scale bars = 50 μm.

and 4B). At low light intensity, EGR1 expression throughout the SCN was significantly blunted in PACAP deficient mice compared to that of wild type mice (Fig 4C and 4D).

## No difference in light induced Egr1 mRNA in the SCN of PACAP deficient and wild type mice at late night

At late subjective night (ZT22) light at both high and low light intensity significantly induced Egr1 mRNA throughout the SCN in both wild type and PACAP deficient mice, and no difference was found between the two genotypes (Fig 5). Similarly, the light induced EGR1 immunoreactivity did not differ during neither high nor low light intensity.

## Discussion

PACAP signaling via the PAC1 receptor has previously been shown to be involved in light entrainment and phase shift of the circadian rhythm in mice [8, 9, 18, 25, 26]. Mice lacking the

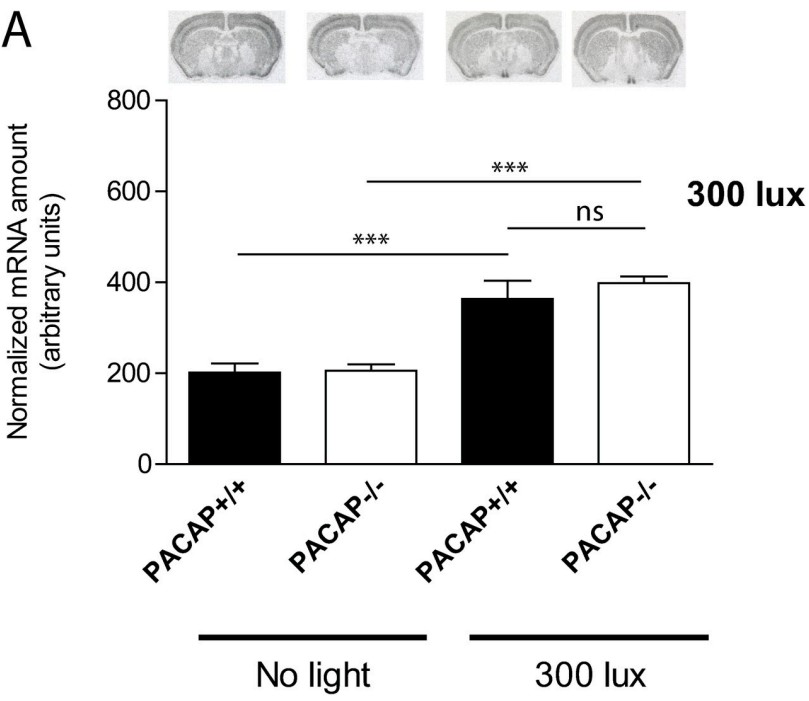

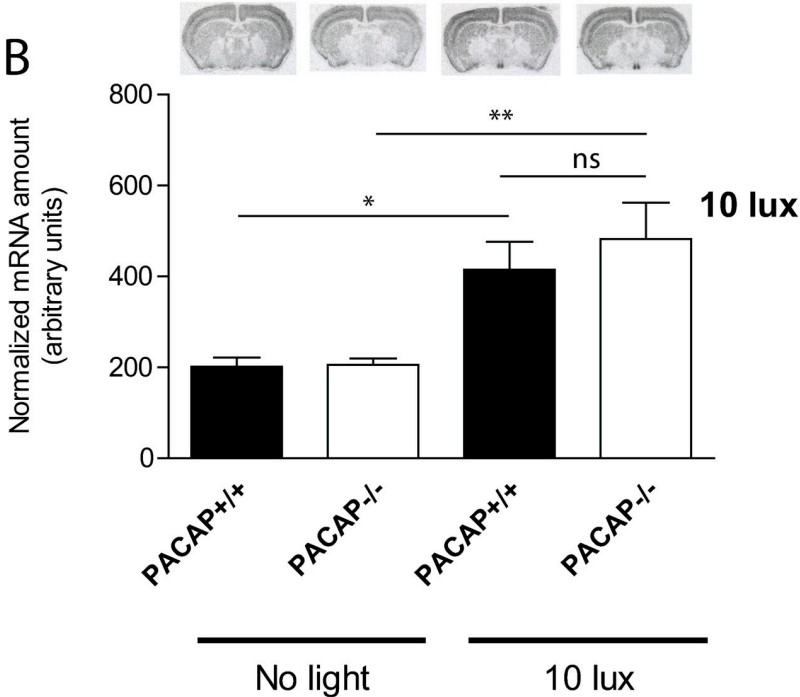

**Fig 5.** EGR1 mRNA (isotop labelled probe) in the SCN in wild type and PACAP deficient mice at late night light (ZT23) stimulated with 300 lux (A) or 10 lux (B) light intensity at ZT22. EGR1 mRNA amount is presented as group mean (± SEM, n = 8 animals) and results of wild type mice are shown in the black bars and PACAP deficient mice in the white bars. *** $p<0.001$, ** $p<0.005$, * $p<0.05$.

PAC1 receptor or PACAP demonstrate smaller phase delay at early night, most likely due to altered light sensitivity with decreasing light intensities, while light mediated phase advances are less compromised at late night [8, 25]. PACAP signaling mechanism involved in altered phase shifting properties during early night involves induction of the expression of *Fos* and the two light responsive genes *Per1* and *Per2*. The induction is reduced in PAC1 mice compared to wild type mice after light stimulation at ZT16-17 [7, 9, 26, 27]. Furthermore, in vitro experiments with SCN brain slices have demonstrated a direct effect of PACAP on *Per1* and *Per2* gene expression in SCN neurons similar to that of light in intact animals [10]. In the present study, we found that PACAP signaling via the RHT is involved in light induction of EGR1 gene expression within neurons of the SCN. This induction was time and light intensity dependent and restricted to the early night at low light intensity. When examining the dynamic of Egr1 expression in the SCN at early night after a 30 min light pulse (300 lux) at ZT16, we found no difference between the two genotypes. Furthermore, these experiments also demonstrated that the highest level of EGR1 mRNA expression occurs 1 h after light stimulation, as reported previously by others [11]. At both early and late night, EGR1 expression is strongly induced by high light intensity and at late night at low light intensity, irrespectively of the presence of PACAP. PACAP is co-stored with glutamate in nerve terminals of the RHT [28, 29] and glutamate is considered to be the primary neurotransmitter of the RHT [6, 30–32]. It is likely that release of glutamate upon light stimulation at night mediates the induction of *Egr1* expression in the SCN neurons since glutamate is known to be a key stimulator of *Egr1* expression via activation of N-methyl-D-aspartate (NMDA) receptors [33]. NMDA stimulation results in elevation of intracellular Ca2+ and light like phase shifts which can be modulated by PACAP signaling [10, 34, 35]. The blunted induction of *Egr1* at low light intensity found in the present study suggests that EGR1 is involved in the modulating effect of PACAP on glutamate/NMDA signaling at early night.

A role of EGR1 in light induced phase shift is, however, not clarified. Although EGR1 is strongly induced by light at night, studies in mice lacking EGR1 revealed normal light entrainment and phase response to bright light during both early and late night [14, 24], and normal expression of FOS after nocturnal light stimulation [14]. In both studies, generalized EGR1 deficient mice were used and neither found difference in phase shifts between the wild type and EGR1 deficient mice stimulated with high light intensities, nor did we find change in light induced *Per1* response in the SCN [24]. Furthermore, we also examined the same mice at low light intensities (10 lux) and found no difference in phase shifts between the genotypes (S1B Fig). It is therefore likely that EGR1 induction in SCN neurons after light stimulation at night is not directly involved in light induced phase shifts (or light entrainment). In this study, generalized PACAP deficient mice were used, and we cannot exclude that our results can be affected by developmental changes in the KO mice.

EGR1 regulates many genes that have diverse cellular functions such as cell proliferation, cell growth, apoptosis, vascular functions, immune response, female reproduction, and learning and memory [36, 37]. Recently, EGR1 was shown to regulate NMDA dependent transcription of the postsynaptic density protein PSD-95 and trafficking of the α-amino-3-hydroxy-5-methyl-4-isoxazole propionic acid receptor (AMPAR) in hippocampal primary neurons [38]. PSD-95 is diurnally expressed in the mouse SCN as other postsynaptic scaffolding proteins [39], and AMPAR play important roles in light induced phase shifts [40]. A role of PACAP as modulator of light NMDA/glutamate mediated neurotransmission via the transcription factor EGR1 in SCN neurons needs further investigation but place PACAP in a potential role shaping synaptic plasticity during the LD cycle and during light stimulation at night.

## Supporting information

**S1 Fig.** (A) *Egr1* mRNA expression during a 24 h LD cycle. Values were analysed using one-anova followed by Bonferroni's Multiple Comparison Test. P < 0.05 was considered statistically significant. Values are given as mean ± SEM (n = 8 in each group, 4 males and 4 females). (B) Light induced phase shifts in wild type (EGR1+/+, black) and EGR1 deficient (EGR1-/-, white) mice during early (ZT16) and late subjective (ZT23) night to a 30 minutes light stimulation (10 lux). Values are given as mean ± SEM (n = 7–13 in each group). NS: not significant using Mann-Whitney U test.
(TIF)

**S1 Material.**
(DOCX)

## Acknowledgments

The skilful technical assistance of Anita Hansen and Tina Wintersø is gratefully acknowledged.

## Author Contributions

**Conceptualization:** Casper Schwartz Riedel, Birgitte Georg, Jan Fahrenkrug, Jens Hannibal.

**Data curation:** Jens Hannibal.

**Formal analysis:** Casper Schwartz Riedel, Jens Hannibal.

**Funding acquisition:** Jens Hannibal.

**Investigation:** Casper Schwartz Riedel, Jens Hannibal.

**Methodology:** Birgitte Georg, Jens Hannibal.

**Project administration:** Jens Hannibal.

**Validation:** Jens Hannibal.

**Visualization:** Jens Hannibal.

**Writing – original draft:** Casper Schwartz Riedel, Birgitte Georg, Jan Fahrenkrug, Jens Hannibal.

**Writing – review & editing:** Birgitte Georg, Jens Hannibal.

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
