## [Decision Letter · Decision Letter 0]

6 Nov 2019

PONE-D-19-28339

Altered light induced EGR1 expression in the SCN of PACAP deficient mice

PLOS ONE

Dear Dr. Hannibal,

Thank you for submitting your manuscript to PLOS ONE. After careful consideration, we feel that it has merit but does not fully meet PLOS ONE’s publication criteria as it currently stands. Therefore, we invite you to submit a revised version of the manuscript that addresses the points raised during the review process.

Your manuscript was reviewed by 3 expert reviewers. All reviewers indicated several very important suggestions. Please address all the issues raised by the reviewers. To fully address the reviewers’ concerns, performing additional experiments is necessary. I strongly encourage you to do so. This letter indicates the due date for your revised manuscript submission. If you need additional time to complete additional experiments, please contact the journal office and notify us of the approximate date you will be submitting your revised manuscript. Please indicate the approximate age of mice you used in this study. Please also indicate detailed procedures on how you handled (perfused) mice in the dark. You have stated that all relevant data are within the manuscript and its Supporting Information files, however, there is no file uploaded. Please upload Supporting Information.

We would appreciate receiving your revised manuscript by Dec 21 2019 11:59PM. To enhance the reproducibility of your results, we recommend that if applicable you deposit your laboratory protocols in protocols.io, where a protocol can be assigned its own identifier (DOI) such that it can be cited independently in the future. For instructions see: http://journals.plos.org/plosone/s/submission-guidelines#loc-laboratory-protocols

We look forward to receiving your revised manuscript.

Kind regards,

Shin Yamazaki, Ph.D.

Academic Editor

PLOS ONE

Journal Requirements:

Reviewers' comments:

Reviewer's Responses to Questions

**Comments to the Author**

1. Is the manuscript technically sound, and do the data support the conclusions?

Reviewer #1: Yes

Reviewer #2: No

Reviewer #3: Partly

2. Has the statistical analysis been performed appropriately and rigorously? 

Reviewer #1: Yes

Reviewer #2: No

Reviewer #3: I Don't Know

3. Have the authors made all data underlying the findings in their manuscript fully available?

Reviewer #1: Yes

Reviewer #2: Yes

Reviewer #3: No

4. Is the manuscript presented in an intelligible fashion and written in standard English?

Reviewer #1: Yes

Reviewer #2: Yes

Reviewer #3: Yes

5. Review Comments to the Author

Reviewer #1: The authors applied two methods to measure egr1 protein and RNA levels independently in SCN in response to light stimulation and concluded that egr1 induction in PACAP deficient mice is blunted only with low light at ZT17. Although the conclusion is solid, additional details will strengthen the results.

1) Only one time point (60min after initiation of light stimulation) was measured in all experiments. Is it possible that egr1 is induced with different dynamics instead of the response being dampened in mutant mice?

2) Although egr1 signal is likely from neuron, no direct evidence in the paper confirmed this. A co-staining with neuron markers such as NeuN will help. In addition, IF in figure 2 does not include all conditions tested in Figure 1 and no quantification was shown.

Reviewer #2: The study reports that PACAP-deficient mice show a significant reduction in SCN EGR-1 expression in response to low-intensity but not high-intensity light pulses. Furthermore, this effect occurs at only one of two phases of the night during light cycles. The Discussion also argues that EGR-1 signaling does not serve a major or direct role in entrainment of the circadian clock to light.

Major concerns:

The report is brief but provides some additional information. The ability to induce SCN EGR-1 by light exposure has been reported previously. PACAP has an established role in the entrainment of the SCN clock to retinal light exposure. The role of EGR-1 in the SCN is not known, although it does serve important functions in hippocampal synaptic plasticity. This study suggests PACAP may not have a critical role in EGR-1 induction by light. It provides an incremental increase in knowledge about the light entrainment pathway. The manuscript discusses other studies in which EGR-1 was not found to play a large role in entrainment, but it does not contribute much additional evidence on this point.

The Kruskal–Wallis or a one-way ANOVA should be used here rather than the Mann-Whitney U-test because comparisons are made between multiple (four) groups. After the groups are compared properly, it will be possible to determine whether significant differences still exist.

Minor concerns:

The source and nature of the PACAP-deficient mice was not stated. Are these complete PACAP knockouts? Do they have any other behavioral or physiological deficits?

What was the age of the mice used?

Although the text was very readable, some words were used incorrectly, for example:

Line 58, classically instead of classical.

Line 110, Lightning instead of light.

Line 150, Fixated instead of fixed.

Reviewer #3: This study assessed the role of PACAP on light-induced EGR1 expression in the SCN. The results showed that PACAP deficiency resulted in blunted EGR1 induction following a 10 lux light pulse at ZT17, but had no effect on EGR1 induction after a 300 lux pulse at ZT17 or 23, or a 10 lux pulse at ZT23. Although the study was straightforward, there are a few limitations in the design of the study, which can be addressed by additional experiments to include more time points of sampling and more detailed analysis of the results.

Major points:

1. Light-induced EGR1 was examined at a single time point following the light pulse. A time course should be established, and it will be more informative to look beyond a single time point to examine how the time course of EGR1 induction was affected by PACAP. For example, although PACAP seemed to have no effect at 60 minutes following a pulse at ZT23, there might be an effect at 30 minute or 90 minutes after the pulse.

2. Give the fact that PACAP is released from RHT terminals to the SCN, it would be more adequate to analyze the EGR1 induction by SCN subregions, i.e. the retinorecipient core region and the shell region of the SCN.

Minor points:

1. what's the light intensity of the housing condition?

2, how the intensity of the light pulse was measured? at cage top level? eye level of the animals?

3, what's the genetic background the animals? what's the rationale of light pulse at ZT17? (maximum phase delay on PRC? or around dead zone? for this strain?)

4. in Figure 2, some double-label to delineate the boundary and subregion of the SCN will be helpful.

5. the authors discussed the results of EGR1 KO, stating that "EGR1 induction in SCN neurons after light stimulation at night is not directly involve din light induced phase shifts". It's possible that in KO animals, some compensatory mechanisms could have occurred, which saved the animals from behavioral deficits, and had normal phase shifts. It could also be the case for PACAP KO. A conditioned KO would be a better model to test the role of these genes.

6. For statistical analysis, a two factor ANOVA seems to be a better choice. Any reason/justification for using Mann-Whitney U test?

6. PLOS authors have the option to publish the peer review history of their article (what does this mean?). If published, this will include your full peer review and any attached files.

Reviewer #1: No

Reviewer #2: No

Reviewer #3: No

---

## [Author Response · Author response to Decision Letter 0]

9 Mar 2020

Reviewer #1: The authors applied two methods to measure egr1 protein and RNA levels independently in SCN in response to light stimulation and concluded that egr1 induction in PACAP deficient mice is blunted only with low light at ZT17. Although the conclusion is solid, additional details will strengthen the results.

1) Only one time point (60 min after initiation of light stimulation) was measured in all experiments. Is it possible that egr1 is induced with different dynamics instead of the response being dampened in mutant mice?

We have included new data in a supplementary file showing the Egr1 expression in wild type and PACAP knockout mice killed 30 min and 180 min after a 30 min light pulse starting at ZT16. These data show that there is no difference in Egr1 induction between the two groups neither before (30 min) nor later (180 min) than the chosen time-point indicating similar dynamics. These data are shown in a new suppl. Fig. 1 (panel A and B).

2) Although egr1 signal is likely from neuron, no direct evidence in the paper confirmed this. A co-staining with neuron markers such as NeuN will help. In addition, IF in figure 2 does not include all conditions tested in Figure 1 and no quantification was shown.

In a manuscript in preparation (Riedel et al.), we address this question. This study investigates the population of neurons expressing EGR1 and whether these neurons co-store FOS. Furthermore, the EGR1 cells are phenotyped to be VIP, AVP, and neuroglobin expressing cells.

Reviewer #2: The study reports that PACAP-deficient mice show a significant reduction in SCN EGR-1 expression in response to low-intensity but not high-intensity light pulses. Furthermore, this effect occurs at only one of two phases of the night during light cycles. The Discussion also argues that EGR-1 signaling does not serve a major or direct role in entrainment of the circadian clock to light.

Major concerns:

The report is brief but provides some additional information. The ability to induce SCN EGR-1 by light exposure has been reported previously. PACAP has an established role in the entrainment of the SCN clock to retinal light exposure. The role of EGR-1 in the SCN is not known, although it does serve important functions in hippocampal synaptic plasticity. This study suggests PACAP may not have a critical role in EGR-1 induction by light. It provides an incremental increase in knowledge about the light entrainment pathway. The manuscript discusses other studies in which EGR-1 was not found to play a large role in entrainment, but it does not contribute much additional evidence on this point.

The Kruskal–Wallis or a one-way ANOVA should be used here rather than the Mann-Whitney U-test because comparisons are made between multiple (four) groups. After the groups are compared properly, it will be possible to determine whether significant differences still exist.

As requested by you and another reviewer, we reanalyze all data using one-way ANOVA followed by Bonferroni’s Multiple Comparison Test. Our revised analysis showed same results as reported in the initial manuscript. We have included all calculations in the supplementary material (statistics).

Minor concerns:

The source and nature of the PACAP-deficient mice was not stated. 

Our strain of PACAP deficient mice was originally provided from Jim Wachek and Chris Colwell (UCLA). Now mentioned on p.4, l. 84. 

Are these complete PACAP knockouts? Yes.

Do they have any other behavioral or physiological deficits? The change in the circadian system was reported by Colwell et al. Am J. Physiol. 2004. 

What was the age of the mice used? 8-12 weeks when included in the study p. 4 l. 83 Although the text was very readable, some words were used incorrectly, for example:

Line 58, classically instead of classical. Has been corrected 

Line 110, Lightning instead of light. Has been corrected

Line 150, Fixated instead of fixed. Has been corrected

Reviewer #3: This study assessed the role of PACAP on light-induced EGR1 expression in the SCN. The results showed that PACAP deficiency resulted in blunted EGR1 induction following a 10 lux light pulse at ZT17 but had no effect on EGR1 induction after a 300 lux pulse at ZT17 or 23, or a 10 lux pulse at ZT23. Although the study was straightforward, there are a few limitations in the design of the study, which can be addressed by additional experiments to include more time points of sampling and more detailed analysis of the results.

Major points:

1. Light-induced EGR1 was examined at a single time point following the light pulse. A time course should be established, and it will be more informative to look beyond a single time point to examine how the time course of EGR1 induction was affected by PACAP. For example, although PACAP seemed to have no effect at 60 minutes following a pulse at ZT23, there might be an effect at 30 minute or 90 minutes after the pulse.

We have included new data in a new supplementary file from wild type and PACAP knockout mice killed 30 min and 180 min after a 30 min light pulse starting at ZT16, which show similar induction in the two groups of animals at all three time-points, suggesting that the dynamic of induction is not changed in the knockout compared to wild type animals.

2. Given the fact that PACAP is released from RHT terminals to the SCN, it would be more adequate to analyze the EGR1 induction by SCN subregions, i.e. the retinorecipient core region and the shell region of the SCN.

EGR1 expression induced after light stimulation occurs in the retinorecipient part of the SCN initial being most intense in the ventrolateral part. We added a new suppl. figure 2, which demonstrate the entire SCN and the area of the SCN where measurements were performed. The data are presented in the manuscript and in supplementary figs and text. 

Minor points:

1. what's the light intensity of the housing condition? During ordinary housing the light intensity is …...for intensities during light stimuli please see revised text p. 5, l. 117-122:

White light was delivered by fluorescent tubes placed on top of the cages. The light intensity could be adjusted from 10-900 lux (measured at the top of the cages) via a resistence. The light intensity was set to 300 and 10 lux measured using an Advantest Optical Power meter TQ8210 (MetricTest, Hayward, CA), with measurements determined at setting of 514 nm; 300 lux (115.0 µW/cm2) and 10 lux (4.3 µW/cm2), respectively.

2, how the intensity of the light pulse was measured? at cage top level? eye level of the animals? See above.

3, what's the genetic background the animals? 129/SV. This is now in the text p. 4, l. 84.

what's the rationale of light pulse at ZT17? (maximum phase delay on PRC? or around dead zone? for this strain?) 

The light pulse was given at ZT16, which is resulting in the maximum phase delay on PRC; Colwell et al. Am J. Physiol. 2004. See above for the rationale of evaluation of EGR1 expression at ZT17.

4. in Figure 2, some double-label to delineate the boundary and subregion of the SCN will be helpful. 

We have added the boundaries of the ventral/central area of retinal projections and the dorsal medial part in a revised fig. 2, panel C (Abrahamson and Moore Brain Res. 2001).

5. the authors discussed the results of EGR1 KO, stating that "EGR1 induction in SCN neurons after light stimulation at night is not directly involved in light induced phase shifts". It's possible that in KO animals, some compensatory mechanisms could have occurred, which saved the animals from behavioral deficits, and had normal phase shifts. It could also be the case for PACAP KO. A conditioned KO would be a better model to test the role of these genes

A conditioned KO would probably reveal whether compensatory mechanisms play a role, this or a model specifically knocking out PACAP in melanopsin RGCs would also be of interest to evaluate the role of PACAP in light entrainment, it is however not available in our laboratory. 

6. For statistical analysis, a two factor ANOVA seems to be a better choice. Any reason/justification for using Mann-Whitney U test?

As requested by you and one other reviewer, we reanalyzed all data using ANOVA followed by Bonferroni’s Multiple Comparison Test. Our revised analysis showed the same results as reported in our initial manuscript. We now include all calculations in the supplementary material (statistics).

---

## [Decision Letter · Decision Letter 1]

6 Apr 2020

PONE-D-19-28339R1

Altered light induced EGR1 expression in the SCN of PACAP deficient mice

PLOS ONE

Dear Dr. Hannibal,

Thank you for submitting your manuscript to PLOS ONE. After careful consideration, we feel that it has merit but does not fully meet PLOS ONE’s publication criteria as it currently stands. Therefore, we invite you to submit a revised version of the manuscript that addresses the points raised during the review process.

Because reviewer #1 wasn’t available at this time, I asked reviewers #2 and #3 to review your revised manuscript. Although reviewer #3 recommended accepting your revised manuscript, the reviewer #2 indicated several suggestions. Please revise the manuscript according to those suggestions. The reviewer #2 also suggested moving the figures in the supporting information to the main text (into the results section as actual figures). I believe this will make it easier to read, so please consider doing so.

We would appreciate receiving your revised manuscript by May 21 2020 11:59PM. To enhance the reproducibility of your results, we recommend that if applicable you deposit your laboratory protocols in protocols.io, where a protocol can be assigned its own identifier (DOI) such that it can be cited independently in the future. For instructions see: http://journals.plos.org/plosone/s/submission-guidelines#loc-laboratory-protocols

We look forward to receiving your revised manuscript.

Kind regards,

Shin Yamazaki, Ph.D.

Section Editor

PLOS ONE

Reviewers' comments:

Reviewer's Responses to Questions

**Comments to the Author**

1. If the authors have adequately addressed your comments raised in a previous round of review and you feel that this manuscript is now acceptable for publication, you may indicate that here to bypass the “Comments to the Author” section, enter your conflict of interest statement in the “Confidential to Editor” section, and submit your "Accept" recommendation.

Reviewer #2: (No Response)

Reviewer #3: All comments have been addressed

2. Is the manuscript technically sound, and do the data support the conclusions?

Reviewer #2: Yes

Reviewer #3: Yes

3. Has the statistical analysis been performed appropriately and rigorously? 

Reviewer #2: Yes

Reviewer #3: Yes

4. Have the authors made all data underlying the findings in their manuscript fully available?

Reviewer #2: Yes

Reviewer #3: Yes

5. Is the manuscript presented in an intelligible fashion and written in standard English?

Reviewer #2: Yes

Reviewer #3: Yes

6. Review Comments to the Author

Reviewer #2: Most suggested changes and concerns have been made or addressed.

As mentioned by another reviewer, there should be a mention that the results could be affected by a possible developmental change in the KO mice that compensated for the lack of PACAP.

Because there are only three figures, the authors should consider moving some of the results in the Supporting Information to the Results section.

The writing needs additional improvement. Throughout the text there was not enough attention to important details needed for clarity such as these:

In the Abstract use 'PACAP-induced" and "light-induced".

In the Introduction:

line 53, Shouldn't "derivates" be "deviates"?

line 66, "(IEGs)" should be "(IEG)".

line 69, Should be "PACAP has previously".

line 109, "fixated" should be "fixed" here, in line 169, and elsewhere.

Describe the ingredients of Stefanini's fixative the first time it's mentioned.

line 134, "digoxiginin" should be "digoxigenin"

line 207, "(p=***)" should be "(p<0.001)"

Are Figures 1 and 2 from digoxigenin or isotopic results?

line 328, In the Figure 2 caption, instead of "Ventral and central retinorecipient and dorsal portion of the mid SCN" it would be more consistent with the cited reference to refer to this as the "core and shell regions" and described as such in the Results.

Reviewer #3: (No Response)

7. PLOS authors have the option to publish the peer review history of their article (what does this mean?). If published, this will include your full peer review and any attached files.

Reviewer #2: No

Reviewer #3: No

---

## [Author Response · Author response to Decision Letter 1]

21 Apr 2020

Dear Academic Editor Shin Yamazaki

Please receive our revised manuscript entitled:

Altered light induced EGR1 expression in the SCN of PACAP deficient mice 

by

Casper Schwartz Riedel, Birgitte Georg, Jan Fahrenkrug and Jens Hannibal

We thank you and the two reviewers for the constructive comments. We have revised our manuscript by addressing all points raised by reviewer #2. A point to point response can be found below. 

We agree that some of the figures present as supplementary figures in the previous version of the manuscript could be placed in the main manuscript. We therefore added two-revised Fig. 1 and Fig.2 (original Suppl. Fig. 1 and parts of Suppl. Fig.2) into the revised version of the manuscript. Furthermore, we addressed all the minor points raised by reviewer #2. 

Our revised manuscript including a track-change version of the revised version and supplementary material, which has been uploaded at Manuscript Central. The manuscript follows the guidelines for manuscripts submitted to PloSOne. The manuscript has not been submitted elsewhere.

Yours sincerely,

Jens Hannibal, Associate Professor, MD., PhD., DMSc.

 

Reviewer #2: 

Most suggested changes and concerns have been made or addressed.

As mentioned by another reviewer, there should be a mention that the results could be affected by a possible developmental change in the KO mice that compensated for the lack of PACAP.

This is now mentioned in the discussion p. 10, l. 283: “In this study, generalized PACAP deficient mice were used, and we cannot exclude that our results can be affected by developmental changes in the KO mice.”

Because there are only three figures, the authors should consider moving some of the results in the Supporting Information to the Results section.

We agree that some of the figures present as supplementary figures in the previous version of the manuscript could be placed in the main manuscript. We therefore added two-revised Fig. 1 and Fig.2 (original Suppl. Fig. 1 and parts of Suppl. Fig.2) into the revised version of the manuscript. 

The writing needs additional improvement. Throughout the text there was not enough attention to important details needed for clarity such as these:

In the Abstract use 'PACAP-induced" and "light-induced".

p.2. l. 32: “we used PACAP deficient mice to evaluate its role in light induced gene expression of EGR1 in SCN neurons during early (ZT17) and late (ZT23) subjective night at both high (300 lux) and low (10 lux) white light intensity”

.

In the Introduction:

line 53, Shouldn't "derivates" be "deviates"? yes, corrected!

line 66, "(IEGs)" should be "(IEG)". yes, corrected!

line 69, Should be "PACAP has previously". yes, corrected!

line 109, "fixated" should be "fixed" here, in line 169, and elsewhere. yes, corrected!

Describe the ingredients of Stefanini's fixative the first time it's mentioned. yes, corrected!

line 134, "digoxiginin" should be "digoxigenin" yes, corrected!

line 207, "(p=***)" should be "(p<0.001)" yes, corrected!

Are Figures 1 and 2 from digoxigenin or isotopic results? The revised Fig. 2 is digoxigenin and the revised fig. 3 and 5 are isotopic. This is now stated in the figure captions.

line 328, In the Figure 2 caption, instead of "Ventral and central retinorecipient and dorsal portion of the mid SCN" it would be more consistent with the cited reference to refer to this as the "core and shell regions" and described as such in the Results. yes, corrected!

---

## [Editor Report · Decision Letter 2]

22 Apr 2020

Altered light induced EGR1 expression in the SCN of PACAP deficient mice

PONE-D-19-28339R2

Dear Dr. Hannibal,

We are pleased to inform you that your manuscript has been judged scientifically suitable for publication and will be formally accepted for publication once it complies with all outstanding technical requirements.

With kind regards,

Shin Yamazaki, Ph.D.

Section Editor

PLOS ONE
---

## [Editor Report · Acceptance letter]

27 Apr 2020

PONE-D-19-28339R2 

Altered light induced EGR1 expression in the SCN of PACAP deficient mice 

Dear Dr. Hannibal:

I am pleased to inform you that your manuscript has been deemed suitable for publication in PLOS ONE. Congratulations! Your manuscript is now with our production department. 

With kind regards,

on behalf of

Dr. Shin Yamazaki 

Section Editor

PLOS ONE